# ACHIEVING LOW-BIT MUON THROUGH SUBSPACE PRESERVATION AND GRID QUANTIZATION

**Huaijin Wu**[1†]**, Bingrui Li**[2†]**, Yebin Yang**[1]**, Yi Tu**[1]**, Zhanpeng Zhou**[1]**,
Jianfei Chen,**[2] **Junchi Yan**[1*]
[1]School of Computer Science & School of Artificial Intelligence, Shanghai Jiao Tong University
[2]Tsinghua university
{whj1201, R0.0O_yyb, zzp1012, yanjunchi}@sjtu.edu.cn
tuyi.sjtu2023@gmail.com  {lbr22, jianfeic}@tsinghua.edu.cn

## ABSTRACT

Training Large Language Models (LLMs) faces severe memory constraints due to the increasing size of model parameters and optimizer states. The Muon optimizer, which is based on matrix orthogonalization, has recently demonstrated significant potential and offers considerable memory advantages over AdamW by utilizing only the first moment. However, how to apply memory-reduction techniques to further compress the optimizer states of Muon remains underexplored. Directly applying existing methods may encounter significant difficulties due to the orthogonalization process. In this work, we investigate the low-bit compression of Muon and systematically analyze the quantization error exacerbated by orthogonalization. We identify that the error primarily originates from the top singular subspace and the outlier patterns of moment matrix appearing across both dimensions. To address this, we propose 4-bit-Muon-GRASP (GRid And Subspace Preserving), which compresses the Muon optimizer states to 4 bits using grid quantization, while preserving the top singular subspace with minimal overhead. We evaluate 4-bit-Muon-GRASP through pre-training on LLaMA-130M, 350M, and 1.1B architectures and fine-tuning on 7B models for various reasoning tasks. Extensive experiment results show that our 4-bit-Muon-GRASP achieves accuracy comparable to full-precision counterparts while reducing training memory consumption by up to 28%. The source code is publicly available at https://github.com/wuhuaijin/lowbit-Muon.

## 1 INTRODUCTION

Large Language Models (LLMs) have shown impressive performance across multiple domains (Yang et al., 2026), including language translation and math reasoning. The growing size of deep learning models has led to significant challenges in terms of memory consumption and computational efficiency (Yuan et al., 2025; Wu et al., 2025), particularly during training (Chowdhery et al., 2023; Rajbhandari et al., 2021). For instance, pre-training a 5B model from scratch using AdamW with only one sequence of length 1024 exceeds the memory capacity of an NVIDIA A100, with the full-precision (fp32) optimizer state alone surpassing 40GB, due to the need for a buffer of 2x the model size to track both the first and second moments.

Existing efforts including GPU sharding (Rajbhandari et al., 2020; Zhao et al., 2023) and CPU offloading (Ren et al., 2021) have focused on reducing the memory consumption of optimizer states at the hardware level. On the other hand, there are two main approaches in the algorithmic domain: factorization uses low-rank approximation to optimize states (Shazeer & Stern, 2018; Zhao et al., 2024), while quantizing the optimizer to lower precision is particularly attractive due to its simplicity and broad applicability. Existing works have successfully compressed the optimizer state to low-bit (8-bit,4-bit), primarily focusing on AdamW and SGD (Wang et al., 2024; Dettmers et al., 2021; Li

---

*Correspondence author. † equal contribution. This work was partly supported by Scientific Research Innovation Capability Support Project for Young Faculty (U40) of the Ministry of Education of China (SRICSPYF-ZY2025019).

et al., 2023). As a result, optimizing the memory usage of the optimizer allows the saved memory to be reallocated for a larger model or an increased batch size.

Recently, Jordan et al. (2024) introduced the Muon optimizer, which incorporates orthonormalized update rules and has demonstrated substantial advantages. Large-scale studies (Liu et al., 2025) report that Muon nearly doubles the efficiency of AdamW, and Muon has been successfully deployed in foundational models such as Kimi-K2 (Team et al., 2025). From the perspective of memory-efficient training, Muon can reduce optimizer state memory usage by approximately 50% compared to the widely-used AdamW in LLM training, as it only requires storage of the first moment. Therefore, further compressing the Muon optimizer state holds significant potential.

However, the application of memory-reduction techniques to Muon remains an open question. Directly applying low-bit compression techniques, such as those used in AdamW (Dettmers et al., 2021; Li et al., 2023), may encounter great challenges due to the orthonormalization process. In this paper, we explore the low-bit compression of Muon optimizer, a problem that, to the best of our knowledge, has not been attempted before. **Our contributions are as follows**:

1) We conduct a systematic analysis of the 4-bit compression error in Muon and find that the Newton-Schulz iteration exacerbates the quantization error, primarily due to the top singular subspace. In light of this, we divide the moment matrix into two parts: the top singular subspace and the residual singular subspace, and compress them separately.

2) We propose 4-bit-Muon-GRASP (GRid And Subspace Preserving) with two key techniques: subspace preservation and grid quantization. Specifically, we suggest using a relatively mild compression (8-bit) to preserve the top singular subspace, with the memory overhead being negligible, while compressing the residual singular subspace to 4-bit. Moreover, given that the outlier pattern of moments appears across both dimensions, we introduce grid quantization to provide more accurate bounds via normalizing both row and column directions.

3) We evaluate our 4-bit-Muon-GRASP through both pre-training and fine-tuning. Specifically, we pre-train on LLaMA-130M, 350M, and 1B architectures with up to 31.5B tokens and fine-tune on 7B models for either general or specific reasoning tasks. The performance of our compressed optimizers is assessed through training curves and downstream tasks. Across all tasks, our 4-bit optimizers achieve accuracy comparable to their full-precision counterparts, while reducing total training memory consumption by up to 28%. To the best of our knowledge, it is the most memory-efficient optimizer among all low-bit optimization methods.

## 2 PRELIMINARIES

### 2.1 MUON OPTIMIZER

The Muon optimizer (Jordan et al., 2024) is a recently proposed novel optimization method, specifically designed for neural network weights representable as matrices. Unlike traditional optimizers, Muon introduces a key innovation by integrating orthogonalization into the moment update process. At iteration $t$, given the current weight $\mathbf{W}_{t-1} \in \mathbb{R}^{n \times m}$, learning rate $\eta_t$, momentum $\mu$, and objective $\mathcal{L}_t$, the update rule for the Muon optimizer can be formulated as follows:

$$\mathbf{M}_t = \mu \mathbf{M}_{t-1} + \nabla \mathcal{L}_t(\mathbf{W}_{t-1}), \tag{1}$$

$$\mathbf{O}_t = \text{Newton-Schulz}_p(\mathbf{M}_t, T), \tag{2}$$

$$\mathbf{W}_t = \mathbf{W}_{t-1} - \eta_t \mathbf{O}_t, \tag{3}$$

where $\mathbf{M}_t$ represents the moment buffer at step $t$, initialized as a zero matrix, $p$ is the degree of the Newton-Schulz (NS) iteration polynomial, and $T$ is the number of iteration steps. The NS iteration aims to approximately orthogonalize the update matrix, which is equivalent to replacing the update with $\mathbf{U}\mathbf{V}^\top$, where $\mathbf{U}\mathbf{\Sigma}\mathbf{V}^\top = \mathbf{M}_t$ is the singular value decomposition (SVD) of $\mathbf{M_t}$. The NS iteration approximation avoids the high computational cost of SVD while still achieving the orthogonalization of the moment matrix, leading to an isomorphic parameter update.

For the NS iteration, we set $\mathbf{X}_0 = \frac{\mathbf{M_t}}{\|\mathbf{M}_t\|_F}$. Following Jordan et al. (2024); Team et al. (2025), where both $p$ and $T$ are set to 5, we denoted this setting as *official choice*. then the update $\mathbf{X}_k$ from $\mathbf{X}_{k-1}$

at each step $k$ is as follows:

$$\mathbf{X}_k = a\mathbf{X}_{k-1} + b\left(\mathbf{X}_{k-1}\mathbf{X}_{k-1}^\top\right)\mathbf{X}_{k-1} + c\left(\mathbf{X}_{k-1}\mathbf{X}_{k-1}^\top\right)^2\mathbf{X}_{k-1} \tag{4}$$

where $a$, $b$, and $c$ are the coefficients. To ensure proper convergence, we tune the coefficients so that the polynomial $f(x) = ax + bx^3 + cx^5$ has a fixed point near 1. In this paper, we follow the official design, using $a = 3.4445$, $b = -4.7750$, and $c = 2.0315$ for LLM training.

**Remark.** A recent work (Liu et al., 2025) has extended Muon from classic models to LLM-scale training by incorporating weight decay and carefully adjusting the per-parameter update scale. As our goal is also for LLM, we adopt these techniques and follow their setting in the training of LLMs.

### 2.2 QUANTIZATION AND DEQUANTIZATION

Quantizing the optimizer states to lower precision is an effective method to compress optimizer states for memory savings. Specifically, the optimizer states $\mathbf{M}_t$ are compressed to $\mathbf{M}_t^q$ using a quantizer at step $t$ and then decompressed with a dequantizer for use at step $t + 1$.

**Quantization.** Quantization is the process of converting full-precision tensors into low-precision formats. Let $X \in \mathbb{R}^p$ represent a full-precision tensor, and let $\text{QUANT}_b$ be a $b$-bit quantizer that reduces $X$ to a discrete value chosen from a set of $2^b$ possible values. The quantization process involves two operations: normalization $\mathcal{N}(\cdot)$ and mapping $\mathcal{M}(\cdot)$, which are applied sequentially and element-wise. Specifically, for each element $x_i \in X$, the quantized value is given by

$$q_i := \text{QUANT}_b(x_i) = \mathcal{M} \circ \mathcal{N}(x_i). \tag{5}$$

Taking signed values as an example, the normalization operator transforms elements of $X$ into the range $[-1, 1]$ according to the following formula:

$$\mathcal{N}(x_i) = \frac{x_i}{\max_{1 \leq j \leq p} |x_j|}, \tag{6}$$

The scaling factors involved in normalization are referred to as *quantization scales*, and they are stored along with the quantized tensor for dequantization. The normalization range determines the granularity of quantization, with common granularities including per-tensor, per-token, per-channel, group-wise, and block-wise.

The mapping operator $\mathcal{M}$ for $x \in \mathbb{R}$ in a $b$-bit quantizer is defined as follows:

$$\mathcal{M}(x) = \arg\min_{j \in \mathbb{T}_b} |x - \mathcal{R}(j)| \tag{7}$$

where $\mathcal{R}$ is the quantization mapping function. The set $\mathbb{T}_b = \{0, 1, \ldots, 2^{b-1}\}$ represents the discrete set of possible values, and the mapping $\mathcal{R}$ is an element-wise function that maps each element of $\mathbb{T}_b$ into the normalized range $[-1, 1]$. Different quantization mappings can be employed, such as linear mapping and dynamic exponent mapping.

**Dequantization.** The dequantizer, denoted as DEQUANT, performs the inverse operation of the quantizer, recovering the approximate original value. This process is defined as follows:

$$\hat{x} = \text{DEQUANT}(q_i) = \mathcal{N}^{-1} \circ \mathcal{R}(q_i) \tag{8}$$

## 3 METHODOLOGY

In this section, we first present the challenge of quantizing the Muon optimizer states to 4 bits and analyze the associated quantization error. We then describe the design of our proposed 4-bit-Muon-GRASP, highlighting two key techniques: top singular subspace preservation and grid quantization.

### 3.1 CHALLENGES

A straightforward way to implement 4-bit Muon is to directly apply group quantization to the moment matrix $\mathbf{M}_t$, as in Dettmers et al. (2021); Li et al. (2023). In this paper, we refer to this naive method as 4-bit-Muon-base.

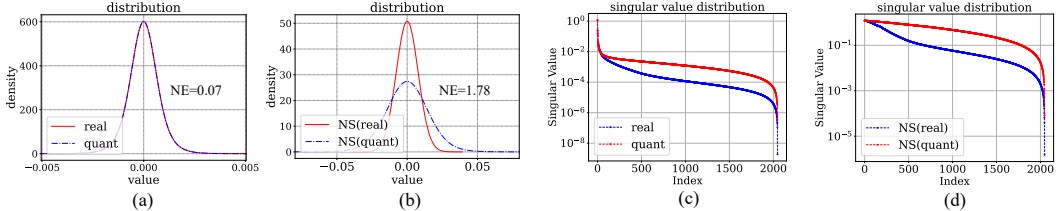

Figure 1: Visualization of momentum in *transformer.layers.7.attn.o_proj* in a LLaMA model. (a) The distribution of matrix (real) and their 4-bit compressions (quant). (b) Distribution of the matrix after NS iteration (NS(real)) and its 4-bit compressions after NS iterations (NS(quant)). (c) and (d): Distribution of singular values of the matrices in (a) and (b), displayed on a $\log_{10}$ scale.

To assess the quantization errors of matrices, we first define the relative error between matrix $\mathbf{A}$ and matrix $\mathbf{B}$ as follows:

$$\mathrm{RE}(\mathbf{A}, \mathbf{B}) = \frac{\|\mathbf{A} - \mathbf{B}\|_F}{\|\mathbf{B}\|_F}. \tag{9}$$

**NS iterations amplify quantization error.** Unlike optimizers such as SGDM (Qian, 1999), Adam (Kingma & Ba, 2014), and AdamW (Loshchilov & Hutter, 2017), which compute updates in an element-wise fashion, the moment matrix in Muon undergoes an orthogonalization step, achieved through Newton-Schulz (NS) iterations. This process may introduce significant quantization errors. To assess whether this process amplifies the quantization error, we visualize the real matrices and their 4-bit compressions. Fig. 1(a) and (b) demonstrate that the distributions of $M_t$ and its 4-bit compressions are nearly identical (RE=0.07), while a significant difference emerges between their distributions after NS iterations (RE=1.78). In other words, the disturbances and errors introduced by quantization become much more pronounced after matrix orthogonalization.

**Error Not Caused by Insufficient Iterations.** To study how NS iterations amplify quantization error, we investigate how quantization errors change with the arguments in NS iterations, including the degree of the polynomial and the number of steps. Fig. 2 shows that as the number iterations and/or the degree of the polynomial increases, although the accuracy of the NS iteration improves, the quantization error even becomes higher. On the other hand, we analyze the singular values of the moment matrix. From Fig. 1(c) and (d), it is evident that the quantization operation consistently increases the singular values before and after NS iterations. Instead, this suggests that the quantized matrix requires fewer NS iterations to converge. Together, the evidences indicate that quantization errors are not due to insufficient iterations.

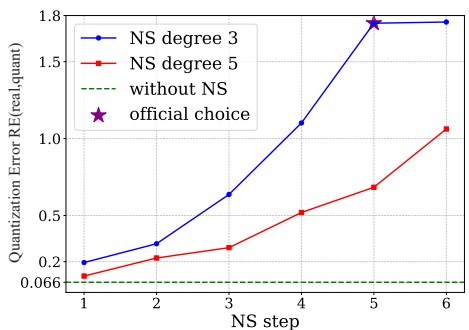

Figure 2: Comparison of quantization error with different arguments in NS iteration.

**Top singular subspace suffers large quantization error.** To further investigate the source of error amplification in the NS iteration, we analyze the moment matrix by dividing it into the top singular space and the residual singular space. Formally, suppose $\mathbf{U}\boldsymbol{\Sigma}\mathbf{V}^\top = \mathbf{M}$ is the SVD of the moment matrix $\mathbf{M} \in \mathbb{R}^{m \times n}$. Then, we define:

$$\mathbf{M}_{\mathrm{top}} := \mathbf{M}_k = \mathbf{U}_k \boldsymbol{\Sigma}_k \mathbf{V}_k^\top, \quad \mathbf{M}_{\mathrm{res}} = \mathbf{M} - \mathbf{M}_{\mathrm{top}}, \tag{10}$$

where $\mathbf{U}_k \in \mathbb{R}^{m \times k}$, $\mathbf{V}_k \in \mathbb{R}^{m \times k}$ are top-$k$ singular vectors and $\boldsymbol{\Sigma}_k \in \mathbb{R}^{k \times k}$ contains top-$k$ singular values. We choose different ranks $k$ to construct $\mathbf{M}_{\mathrm{top}}$ and $\mathbf{M}_{\mathrm{res}}$, and compress them to 4-bit, resulting in $\hat{\mathbf{M}}_{\mathrm{top}}$ and $\hat{\mathbf{M}}_{\mathrm{res}}$, and compute the quantization error before and after NS iterations.

Table 1: Quantization error before and after NS iterations. The results represent the average values obtained across all parameters during the first 100 training iterations of the 1.1B LLaMA model.

| $k$ | RE ($\mathbf{M}_{\mathrm{top}}$) | RE ($\mathbf{M}_{\mathrm{res}}$) |
|-----|------------------|------------------|
| 64  | $0.08 \rightarrow 3.31$ | $0.09 \rightarrow 0.47$ |
| 128 | $0.08 \rightarrow 2.42$ | $0.09 \rightarrow 0.63$ |
| 256 | $0.08 \rightarrow 1.76$ | $0.09 \rightarrow 0.35$ |
| 512 | $0.08 \rightarrow 1.26$ | $0.09 \rightarrow 0.42$ |

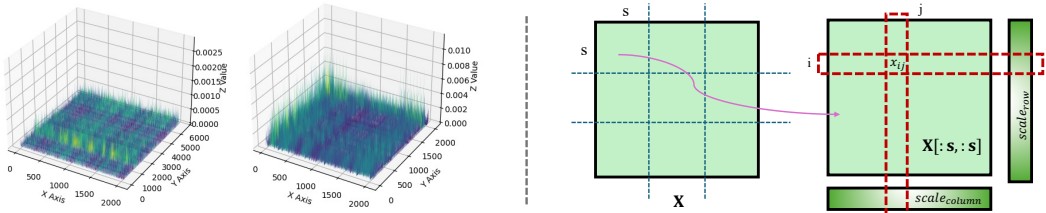

Figure 3: Left: Outlier patterns of moment tensor. Right: Illustration of grid quantization.

As shown in Tab. 1, we find that the quantization errors of the top singular spaces and residual singular spaces are comparable before NS iteration. However, after NS iteration, the difference between the two becomes significant. Specifically, when setting $k = 64$ for all parameter tensors in the model, NS iteration amplifies the quantization error of $\mathbf{M}_{\text{top}}$ by $40\times$, while increasing it only $5\times$ for the residual matrix $\mathbf{M}_{\text{res}}$. This indicates that NS iteration primarily amplifies quantization errors in the top singular subspaces, suggesting that quantization methods can be designed separately for different subspaces.

## 3.2 SUBSPACE PRESERVING

Given that the top singular subspace incurs significant quantization error in Muon, we propose using relatively mild compression to better preserve the information contained in the top singular subspace. Additionally, the NS iteration amplifies all singular values, leading to the magnification of even originally small values to an extent that cannot be overlooked. As a result, relying solely on the top singular subspace fails to capture all the information in the moment matrix, leading to accuracy loss. To address this, we retain the residual subspace of the matrix and apply 4-bit quantization to it.

Regarding how to obtain the top singular space of the moment matrix, using Eq. 10 via SVD is computationally expensive. In this work, we employ a numerical iterative approximation method known as *Power Iteration* to get the top singular vectors, which has also been utilized in Vogels et al. (2019); Ahn et al. (2025). Following them, we warm-start the power iteration using the results from the previous optimizer, so that a single iteration is sufficient to accurately capture the top singular subspace of $\mathbf{M_t}$. Specifically, at each step, we first compute the moment $\mathbf{M}_t = \mu \mathbf{M}_{t-1} + \nabla \mathcal{L}_t(W_{t-1})$, and then use *Power Iteration* to compute the top singular vectors $\mathbf{P}_t, \mathbf{R}_t$ with rank $k$:

$$\mathbf{P}_t \mathbf{R}_t^\top \approx \mathbf{M}_{\text{top}}, \quad \text{where } \mathbf{P}_t \in \mathbb{R}^{m \times k}, \mathbf{R}_t \in \mathbb{R}^{n \times k}. \tag{11}$$

It should be noticed that we perform column normalization on $\mathbf{R}_{t-1}$ to obtain $\mathbf{Q}_t$, and use $\mathbf{Q}_t$ to perform a single step of power iteration. Since $\mathbf{M}_t$ and $\mathbf{M}_{t-1}$ exhibit a certain degree of similarity, we can benefit from reusing $\mathbf{R}_{t-1}$ as the starting point.

We then get the residual singular subspace of $M$ as described in Eq. 10 and apply 4-bit quantization to it. To preserve more information of the top singular subspace, we utilize a relatively mild compression (8-bit) to $\mathbf{P}_t, \mathbf{R}_t$. We store 8-bit $\mathbf{P}_t, \mathbf{R}_t$, and the 4-bit $\mathbf{M}_{\text{res}}$ in the optimizer buffer. Since the rank $k * n + k * m \ll m * n$, the memory overhead of storing $\mathbf{P}_t, \mathbf{R}_t$ is relatively small.

## 3.3 GRID QUANTIZATION

Fig. 3 (a) shows moment tensors in Muon, and we observe that the outlier pattern appears across both dimensions. As a result, neither per-channel group nor per-token group quantization can fully capture such outliers. To achieve higher precision, we propose grid quantization, which performs normalization in both row and column directions to obtain a more precise bound for each entry.

Specifically, for matrix $\mathbf{X}$, we divide it into several blocks of size $s \times s$ ($s$ is the group size), and the element within the block are denoted as $\{x_{i,j} | r_1 \leq i \leq r_2, c_1 \leq j \leq c_2\}$. Then, the row and column quantization scales of this block are formulated as follows:

$$\text{scale}_{r_i} = \max_{r_1 \leq j \leq r_2} x_{i,j}, \quad \text{scale}_{c_j} = \max_{c_1 \leq i \leq c_2} x_{i,j}. \tag{12}$$

---

**Algorithm 1** 4bit-Muon-GRASP

---

**Require:** Weight $\mathbf{W}$, objective $\mathcal{L}$, learning rate $\eta$, momentum $\mu$, weight decay $\lambda$, rank $k$, quantizer QUANT and dequantizer DEQUANT.

1: **repeat**
2:     **if** $t$ is 0 **then**
3:         Randomly initialize right factor $\mathbf{Q}_0 \in \mathbb{R}^{m \times n}$, $\mathbf{M}_0 \leftarrow 0 \in \mathbb{R}^{m \times n}$
4:     **else**
5:         $\mathbf{M}_{\text{res,t-1}} \leftarrow \text{DEQUANT}(\mathbf{M}_{\text{res,t-1}}^q) \in \mathbb{R}^{m \times n}$
6:         $\mathbf{P}_{t-1} \leftarrow \text{DEQUANT}(\mathbf{P}_{t-1}^q) \in \mathbb{R}^{m \times k}$, $\mathbf{R}_{t-1} \leftarrow \text{DEQUANT}(\mathbf{R}_{t-1}^q) \in \mathbb{R}^{n \times k}$
7:         $\mathbf{M}_{t-1} \leftarrow \mathbf{M}_{\text{res,t-1}} + \mathbf{P}_{t-1}\mathbf{R}_{t-1}^\top \in \mathbb{R}^{m \times n}$
8:         $\mathbf{Q}_t \leftarrow \text{ColumnNormalize}(\mathbf{R}_t) \in \mathbb{R}^{n \times k}$
9:     **end if**
10:    $\mathbf{M}_t \leftarrow \mu\mathbf{M}_{t-1} + \nabla\mathcal{L}_t(\mathbf{W}_{t-1}) \in \mathbb{R}^{m \times n}$
11:    $\mathbf{P}_t, \mathbf{R}_t \leftarrow \text{PowerIter}(\mathbf{M}_t, \mathbf{Q}_t) \in \mathbb{R}^{m \times k}, \mathbb{R}^{n \times k}$         ▷ get top singular vectors
12:    $\mathbf{M}_{\text{res,t}} \leftarrow \mathbf{M}_t - \mathbf{P}_t\mathbf{R}_t^\top \in \mathbb{R}^{m \times n}$         ▷ get residual subspace of $\mathbf{M}$
13:    $\mathbf{M}_{\text{res,t}}^q \leftarrow \text{QUANT}_4(\mathbf{M}_{\text{res,t}}) \in \mathbb{R}^{m \times n}$         ▷ 4-bit quantization
14:    $\mathbf{P}_t^q \leftarrow \text{QUANT}_8(\mathbf{P}_{t-1}) \in \mathbb{R}^{m \times k}$, $\mathbf{R}_t^q \leftarrow \text{QUANT}_8(\mathbf{R}_t) \in \mathbb{R}^{n \times k}$     ▷ 8-bit quantization
15:    $\mathbf{O}_t \leftarrow \text{Orthogonalize}(\mathbf{M}_t) \in \mathbb{R}^{m \times n}$         ▷ using newton-schulz iterations
16:    $\mathbf{W}_t \leftarrow \mathbf{W}_{t-1} - \eta_t(\mathbf{O}_t + \lambda\mathbf{W}_{t-1}) \in \mathbb{R}^{m \times n}$
17: **until** convergence criteria met

---

1: **function** POWERITER($\mathbf{B}, \mathbf{Q}$) $\in \mathbb{R}^{m \times n}, \mathbb{R}^{n \times k}$         ▷ single power iteration (from $Q$)
2:     $\mathbf{P} \leftarrow \mathbf{BQ} \in \mathbb{R}^{m \times k}$
3:     $\mathbf{P} \leftarrow \text{Orthogonalize}(\mathbf{P}) \in \mathbb{R}^{m \times k}$         ▷ using QR decomposition
4:     $\mathbf{R} \leftarrow \mathbf{B}^\top\mathbf{P} \in \mathbb{R}^{n \times k}$
5:     **return** $\mathbf{P}, \mathbf{R}$         ▷ $P$ is orthonormal
6: **end function**

---

Then the normalization of grid quantization $x_{i,j}$ can be formulated as:

$$\mathcal{N}_{\text{grid}}(x_{i,j}) = \frac{x_{i,j}}{\min(\text{scale}_{r_i}, \text{scale}_{c_j})} \tag{13}$$

Grid quantization utilizes information in a more fine-grained manner and provides element-wise unique quantization scales, effectively addressing the outliers that appear across both dimensions. Although grid quantization requires storing twice the number of quantization scales compared to group-wise quantization, the resulting memory overhead is negligible.

### 3.4 OVERALL ALGORITHM

By employing the techniques of top subspace preserving and grid quantization, we achieve a lower quantization error for the Muon optimizer (NE=1.78 → NE=0.14). The comparison between NS(real) and NS(quant) is presented in Fig. 4, where the rank of the top subspace is set to 1/16 of the original matrix rank. Algo. 1 outlines the overall procedure of 4-bit-Muon-GRASP, including the detailed steps of Power Iteration.

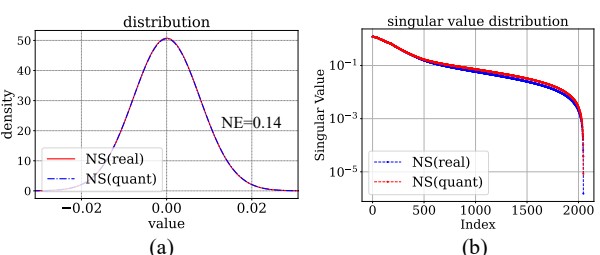

Figure 4: The comparision of NS(real) and NS(quant).

## 4 EXPERIMENTS

We evaluate 4-bit-muon-base and 4-bit-muon-GRASP on both pre-training and fine-tuning of LLMs. We compare our 4-bit muon optimizers with their full-precision counterparts, namely fp32-muon.

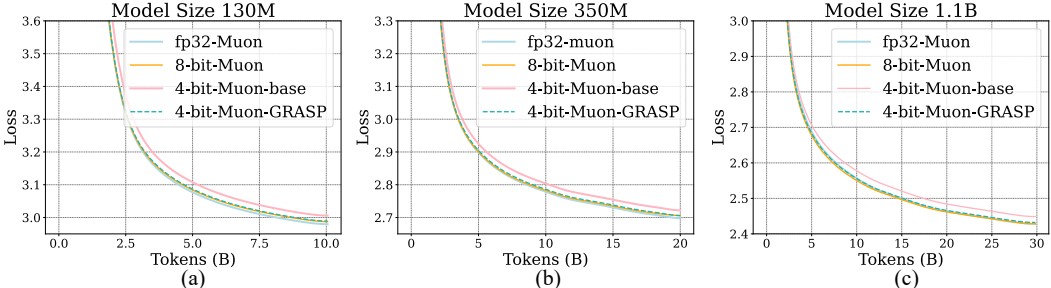

Figure 5: Validation loss comparison of pretraining with different optimizers on Slimpajama.

Table 2: Evaluation of models pre-trained with three optimizers, across downstream tasks for different model sizes. The results are the average of multiple random seeds.

| Model | Optimizer | HellaSwag | ARC-c | ARC-e | boolQ | OBQA | PIQA | SciQ | Avg |
|-------|-----------|-----------|-------|-------|-------|------|------|------|-----|
| 130M | fp32-Muon | 28.4 | 21.8 | 34.3 | 62.1 | 25.8 | 58.3 | 62.2 | 41.8 |
| | 8bit-Muon | 28.4 | 21.6 | 34.1 | 62.2 | 26.9 | 58.5 | 61.7 | 41.6 |
| | 4bit-Muon-base | 28.2 | 20.8 | 34.0 | 62.2 | 27.8 | 58.8 | 59.6 | 41.9 |
| | 4bit-Muon-GRASP | 28.7 | 21.6 | 34.2 | 62.1 | 28.0 | 58.1 | 60.4 | 41.9 |
| 350M | fp32-Muon | 32.4 | 23.5 | 38.3 | 59.4 | 28.8 | 62.0 | 68.0 | 44.6 |
| | 8bit-Muon | 32.5 | 22.3 | 38.1 | 61.3 | 28.4 | 61.8 | 66.5 | 44.5 |
| | 4bit-Muon-base | 31.6 | 22.4 | 37.7 | 61.9 | 26.2 | 61.8 | 64.4 | 43.7 |
| | 4bit-Muon-GRASP | 32.4 | 23.0 | 38.5 | 61.2 | 28.2 | 61.4 | 66.6 | 44.5 |
| 1.1B | fp32-Muon | 40.6 | 25.4 | 42.8 | 60.4 | 30.2 | 66.5 | 69.5 | 48.0 |
| | 8bit-Muon | 40.5 | 25.2 | 42.5 | 60.4 | 30.2 | 66.8 | 71.4 | 48.2 |
| | 4bit-Muon-base | 39.8 | 24.2 | 41.5 | 61.0 | 30.4 | 66.6 | 69.7 | 47.6 |
| | 4bit-Muon-GRASP | 40.4 | 24.8 | 42.3 | 60.5 | 30.6 | 67.4 | 71.3 | 48.2 |

Here we further include 8-bit-Muon as a baseline, which is implemented through group quantization. All experiments run on NVIDIA A100 GPUs.

Following Liu et al. (2025), we employ the Muon optimizer for updating matrix parameters, while RMSNorm, the LM head, and embedding parameters remain optimized using AdamW. Except for the ablation study, the rank of the top singular subspace is set to 1/16 of the original, and we use the widely adopted INT4 and INT8 format for simplicity and efficiency. We implement the quantization-related code using OpenAI Triton kernel to enhance efficiency and achieve real memory reduction. Both the group size and grid size of quantization are set to 128.

## 4.1 PRETRAINING

**Datasets, architecture and hyperparameters.** Following Zhang et al. (2024b), we utilize Slimpajama (Soboleva et al., 2023) as the pre-training dataset and adopt a LLaMA-based architecture with RMSNorm (Zhang & Sennrich, 2019) and SwiGLU activations (Shazeer, 2020). We evaluate three model sizes: 130M, 350M, and 1.1B, and all experiments are conducted with BF16 mixed-precision training to enhance training efficiency. For each model size, we first tune the learning rate for the fp32-Muon from the set $\{2e-3, 1e-3, 6e-4, 3e-4\}$, selecting the best learning rate based on the validation perplexity, respectively. We then apply the same learning rate to the 4-bit-Muon-base and 4-bit-Muon-GRASP models to ensure a fair comparison. See Appendix A for more detailed hyperparameter settings.

**Training Curve.** Fig. 5 illustrates the training curves of different optimizers across three model sizes. The results show that even the 4-bit-Muon-base optimizer does not significantly affect training convergence, with an accuracy loss of 1%. Our proposed 4-bit-Muon-GRASP reduces the training gap with fp32-Muon to less than 0.2%, and on the 1.1B model, 4-bit-Muon-GRASP even achieves no loss in training accuracy.

**Accuracy of 4-bit Optimizers.** In line with Xiong et al. (2025), to assess whether our memory efficient 4-bit optimizers could maintain accuracy on downstream tasks, we evaluate zero-shot performance using the lm-evaluation-harness (Gao et al., 2021) codebase on standard benchmarks, including HellaSwag (Zellers et al., 2019), ARC (Yadav et al., 2019), BoolQ (Clark et al., 2019),

Table 3: Statistics of different optimizers: the step time (s), total memory usage (GB), and validation perplexity (↓) after training for 10K steps.

| | 130M | | | 350M | | | 1.1B | | |
|---|---|---|---|---|---|---|---|---|---|
| | time | mem | PPL | time | mem | PPL | time | mem | PPL |
| **fp32-Muon** | 31.8 | 1.76 | 19.21 | 45.8 | 4.04 | 15.57 | 61.3 | 13.22 | 12.48 |
| **8-bit-Muon** | 31.9 | 1.47 | 19.29 | 45.7 | 3.35 | 15.65 | 61.5 | 13.22 | 12.46 |
| **4-bit-Muon-base** | 31.9 | 1.42 | 19.73 | 45.8 | 3.23 | 15.96 | 61.5 | 10.54 | 12.76 |
| **4-bit-Muon-GRASP** | 32.0 | 1.43 | 19.35 | 46.0 | 3.26 | 15.67 | 61.9 | 10.14 | 12.48 |

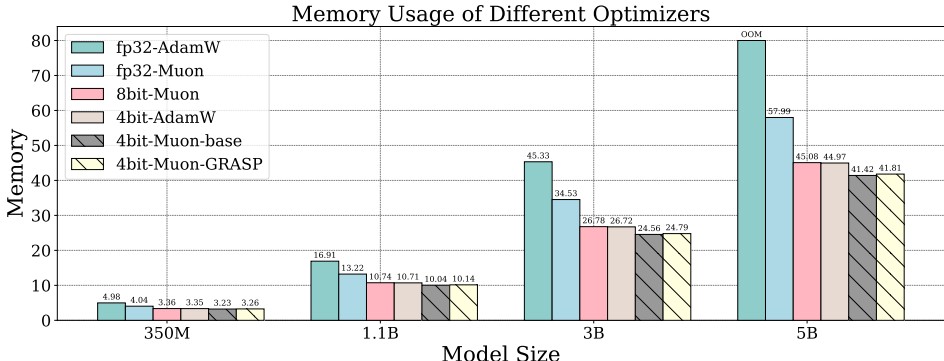

Figure 6: **Total training memory usage** across different optimizers and model sizes, evaluated with one sequence of length 1024 on a single device.

OpenbookQA (Mihaylov et al., 2018), SciQ (Johannes Welbl, 2017), PIQA (Bisk et al., 2020), Winogrande (Sakaguchi et al., 2021). The corresponding results are shown in Tab. 2. For the 350MB model, 4-bit-Muon-base and 4-bit-Muon-GRASP achieve an average accuracy of 44.6 and 44.5, respectively, while the baseline performs at 43.7. A similar trend is observed with the 130M and 1.1B models. These results show that 8-bit-Muon and our 4-bit-Muon-GRASP could match or exceed fp32-Muon performance across all tasks, while 4-bit-Muon-base shows a slight accuracy loss.

**Memory and Computing Efficiency.** The training statistics are presented in Tab. 3, where the 4-bit-Muon-base shows degradation in training accuracy, whereas our 4-bit-Muon-GRASP demonstrates negligible accuracy loss, particularly with the 1.1B model size. Moreover, we find that the training time overhead introduced by quantization is minimal and a detailed breakdown of the cost of power iteration and other optimzier logics are shown in Appendix B.2. Fig. 6 provides a further total memory usage comparison with the full-precision AdamW, 4-bit AdamW, and 8-bit Muon optimizers, with model sizes scaled up to 3B and 5B. We observe that the 4-bit Muon optimizer is the most memory-efficient among all, achieving memory reductions of up to 48% and 28% compared to fp32-AdamW and fp32-Muon, respectively. However, for smaller model sizes, the memory reduction is not substantial, and the memory savings plateau as the bitwidth decreases. This is because we report total memory consumption, encompassing data, activations, weight gradients, and memory fragments, rather than isolating the optimizer's memory consumption alone. We also compare our low-bit compression optimizer with cpu-offloading optimizer, see Appendix B.1 for details.

## 4.2 FINE-TUNING

**Experimental Setup.** We select two pretrained models: the general model Qwen2.5-7B (Yang et al., 2025a) and the domain-specific model Qwen2.5-7B-math (Yang et al., 2024a), to evaluate the effects of different optimizers on both general capabilities and advanced mathematical reasoning.

Our implementation builds upon the verl framework (Sheng et al., 2025). Since Muon requires the full gradient matrix to calculate the updates, and PyTorch Fully Sharded Data Parallel is not directly applicable to Muon, we refer to the public implementation of distributed Muon (hor, 2025; Ahn et al., 2025). We further implement the distributed 4-bit-Muon-base and distributed 4-bit-Muon-

Table 4: Comparison of three optimizers applied to the SFT of the Qwen2.5-7B and Qwen2.5-7B-Math pretrained models.

| Benchmark (Metric) | # Shots | Origin | SFT(fp32) | SFT(4bit-base) | SFT(4bit-GRASP) |
|---|---|---|---|---|---|
| Pretrained model: Qwen2.5-7B | | | | | |
| MMLU(EM) | 0-shot(CoT) | 71.9 | 72.0 | 72.1 | 72.0 |
| HumanEval (Pass@1) | 0-shot | 56.1 | 76.2 | 76.6 | 76.7 |
| MBPP (Pass@1) | 0-shot | 64.4 | 71.4 | 70.3 | 70.9 |
| GSM8K (EM) | 5-shot | 80.3 | 85.4 | 84.8 | 85.2 |
| Pretrained model: Qwen2.5-7B-Math | | | | | |
| MATH (Pass@1) | 0-shot | 65.4 | 70.5 | 69.8 | 70.8 |
| Minerva Math (Pass@1) | 0-shot | 12.1 | 26.8 | 27.9 | 29.0 |
| Olympiad Bench (Pass@1) | 0-shot | 27.3 | 36.0 | 35.9 | 35.4 |
| Average | - | 53.9 | 62.6 | 62.5 | 62.8 |

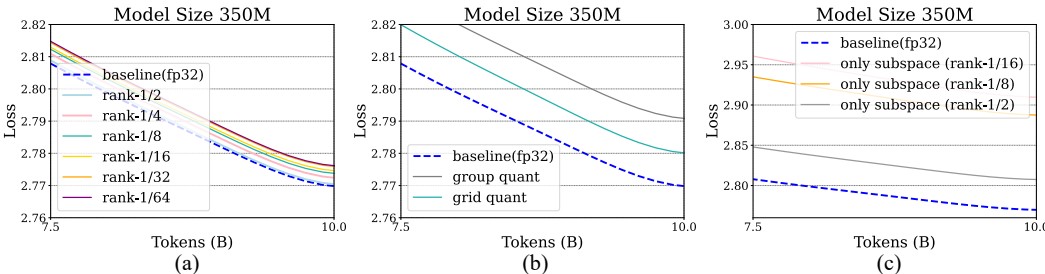

Figure 7: Ablation studies on pretraining: (a) Selection of different top singular space ranks, (b) Comparison of group and grid quantization, (c) Preservation of only the top singular space.

GRASP, where the quantization and dequantization are performed in a partitioned shape, and the subspace preservation of momentum is carried out globally. See Appendix A for training details.

**Performance on Downstream Tasks.** For general model, we use lm-eval-harness (Gao et al., 2021) to evaluate on benchmarks including MMLU (Hendrycks et al., 2020), HumanEval (Chen et al., 2021), MBPP (Austin et al., 2021) and GSM8K (Cobbe et al., 2021). For high-level mathematical reasoning model, we evaluate on established benchmarks including Math (Hendrycks et al., 2021), Minerva Math (Lewkowycz et al., 2022), and Olympiad Bench (olympiad problems 2024, 2024). The evaluation metric and results are shown in Tab. 4, where we compare the performance of fine-tuned models with different optimizers. The results demonstrate that our 4-bit-Muon optimizers will not destroy the capabilities of pretrained models, and the 4-bit-Muon-GRASP achieves performance comparable to that of the 32-bit Muon across all tasks.

## 4.3 ABLATION STUDIES

**How does the rank of the top singular space affect convergence, memory, and computing efficiency?** We preserve the top singular space with different ranks, ranging from 1/64 to 1/2, and perform pre-training on LLaMA-350M. Fig. 7 (a) illustrates that the gap of the training curve between our method and fp32 baseline widens as the rank decreases. Notably, when the rank of the top singular space is set to half of the rank of the matrix $M$, the training loss shows no difference from the baseline. The impact of different ranks on memory usage and time is discussed in Appendix B.4.

**What happens if we preserve only the top singular space while discarding the residual singular space?** Fig. 7 (c) shows the results when only the full-precision top singular space is preserved, with the rank ranging from 1/2 to 1/16. The results demonstrate that discarding the residual singular space leads to a significant degradation in training accuracy. Even with a 1/2 rank approximation, the training accuracy loss exceeds 2%. This underscores the importance of the residual singular space and suggests that the orthogonalization of Muon prevents a straightforward low-rank approximation.

**Comparison between grid quantization and group quantization.** We compare the effects of grid quantization and group quantization on training performance. To provide a clearer comparison, we

directly compress the moment matrix without preserving the top singular space. As shown in Fig. 7 (b), grid quantization reduces the accuracy loss of group quantization by half. See Appendix B.4 for more ablation studies.

**Comparison of different number of *Power Iteration* steps.** To evaluate the approximation accuracy of *Power Iteration* with different step, Fig. 8 report the relative error during training, with 1-step, 2-step, and 3-step power iterations, respectively. The relative error here defined as $RE(\mathbf{U}_k\mathbf{\Sigma}_k\mathbf{V}_k^\top, \mathbf{PR}^\top)$, where $\mathbf{U}_k\mathbf{\Sigma}_k\mathbf{V}_k^\top$ represents the accurate top singular space, and $\mathbf{PR}^\top$ is the approximation obtained through power iteration. The results demonstrate that the error gap between the 1-step and multi-step iterations is minimal. In fact, the approximation error of the 1-step power method already reaches as low as 0.01, indicating that a single iteration is sufficient for the algorithm to accurately identify the top singular vector.

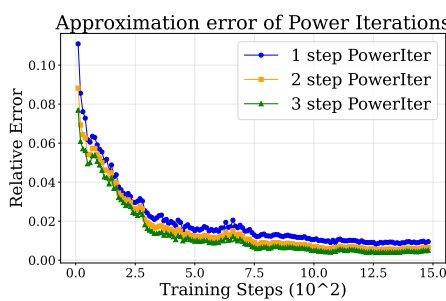

Figure 8: Approximation error during training.

## 5 RELATED WORKS

**Quantization-based memory efficient optimizers.** Dettmers et al. (2021) propose block-wise dynamic quantization, allowing first-order optimizers to operate with 8-bit states, while Li et al. (2023) further compresses the Adam/AdamW optimizer states to 4 bits by applying finer-grained quantization and removing zero points from the second moment. Moreover, Wang et al. (2024) introduces 4-bit second-order optimizers and exemplifies by 4-bit Shampoo.

**Other memory efficient techniques.** Several works have explored approximating gradient statistics with sublinear memory cost relative to the number of parameters. For instance, Adafactor (Shazeer & Stern, 2018) uses the outer product of two vectors to approximate Adam's second moment. SM3 (Anil et al., 2019) approximates the second moment in Adam using the statistics of its covers. LoRA (Hu et al., 2022) freezes the pretrained weights and tunes only the newly initialized low-rank parameters. Additionally, some approaches focus on reducing the memory consumption of activations, such as activation-compressed training and gradient checkpointing, which can be integrated with our optimizer to achieve further memory savings.

## 6 CONCLUSION AND OUTLOOK

In this paper, we introduce 4-bit-Muon-GRASP, a method for compressing the Muon optimizer to improve memory efficiency. By dividing the moment matrix into two parts and applying grid quantization, we are able to reduce memory usage by up to 28% while maintaining performance comparable to full-precision optimizers.

**Limitations and Future Works.** The optimal quantization settings are likely dependent on the task, datasets, and training details, but the exploration in this paper is relatively limited to common LLM training scenarios. Due to resource limitations, our evaluation is currently limited to pretraining on 1.1B models. We identify several open problems for 4-bit-Muon-GRASP, which include: 1) providing strategies or guidelines for automatic rank selection; 2) further enhancing memory efficiency by employing activation reduction methods; and 3) optimizing the efficiency and communication of low-bit optimizer algorithms in distributed scenarios.

## DECLARATION OF AI USE

We used Gemini/ChatGPT to assist in writing:
1) Correcting grammar, improving clarity, and refining the flow of sentences.

The LLMs do not contribute to research ideation, methodology, experimental design, data analysis, interpretation of results, or the creation of substantive academic content or references. We carefully review and approve all suggestions from the models, and we take full responsibility for the final manuscript.

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

# A    DETAILS OF PRE-TRAINING SETTING

Table 5: Hyperparameters for LLaMA model pretraining.

|  | **Parameters** | **130M** | **350M** | **1.1B** |
|---|---|---|---|---|
| Training | lr-schedule | WSD (Wen et al., 2024) | WSD | WSD |
|  | max, min lr | (1e-3, 1e-4) | (6e-4, 6e-5) | (6e-4, 6e-5) |
|  | warmup-ratio | 0.1 | 0.1 | 0.1 |
|  | decay-ratio | 0.99 | 0.99 | 0.99 |
|  | AdamW-$\beta$ | (0.95, 0.9) | (0.95, 0.9) | (0.95, 0.9) |
|  | weight-decay | 0.1 | 0.1 | 0.1 |
|  | grad_clip | 1.0 | 1.0 | 1.0 |
| Model | hidden dim. | 768 | 1024 | 2048 |
|  | #layers | 12 | 22 | 22 |
|  | #q heads | 12 | 16 | 32 |
|  | #kv heads | 4 | 4 | 4 |
|  | context-length | 1024 | 1024 | 1024 |
|  | FFN size | 1024 | 2560 | 5632 |
| Data | tokenzier | LLaMA-2 | LLaMA-2 | LLaMA-2 |
|  | #steps(B) | 10K | 20K | 30K |
|  | #Tokens(B) | 10.5 | 21.0 | 31.4 |
|  | Batch size | 1024 | 1024 | 1024 |

**Pre-training.** We provide details of the LLaMA architecture and the hyperparameters used for pre-training. Tab. 5 presents the key hyperparameters for LLaMA models across different sizes. In all experiments, we use a WSD (warmup-stable-decay) learning rate schedule. The learning rate is warmed up for the first 10% of the training steps and then decays to 10% of the initial value during the final 1% of the training steps.

**Fine-tuning.** For Qwen2.5-7B, we fine-tune the model using the open-source tulu-3-sft-mixture dataset (Lambert et al., 2024). Following Liu et al. (2025), the dataset is packed with a sequence length of 8k tokens, and the learning rate follows a cosine decay schedule, starting from $5 \times 10^{-5}$ and gradually decaying to $2 \times 10^{-6}$. For Qwen2.5-7B-Math, we adopt the NuminaMath CoT dataset (Li et al., 2024) for fine-tuning, which consists of approximately 860,000 mathematical problems paired with their corresponding solutions. In accordance with Qin & Springenberg (2025), we randomly sample 100K instances from the dataset for training, and the dataset is packed with a sequence length of 2k tokens.

# B    MORE EXPERIMENT RESULTS

## B.1    COMPARISON WITH CPU-OFFLOADING OPTIMIZERS

Table 6: The optimizer updating time(s) of different methods.

| Model Size | 130M | 350M | 1.1B | 3B | 5B |
|---|---|---|---|---|---|
| fp32-Muon | 0.18 | 0.25 | 0.39 | 0.55 | 0.75 |
| fp32-Muon-CPU | 0.73 | 0.99 | 3.89 | 11.32 | 17.93 |
| 4-bit-Muon-GRASP (rank 1/4) | 0.55 | 0.91 | 1.82 | 2.85 | 4.21 |
| 4-bit-Muon-GRASP (rank 1/16) | 0.41 | 0.61 | 0.96 | 1.21 | 1.93 |
| 4-bit-Muon-GRASP (rank 1/64) | 0.39 | 0.54 | 0.76 | 0.88 | 1.43 |

Tab. 6 and Tab. 7 shows the optimizer updating times (s) and optimizer memory usage (GB) of different methods, which demonstrate that the optimizer step time for CPU-offloaded Muon increases significantly, especially as the model size grows. For a 5B model, the optimizer time becomes 20 times longer, introducing a substantial overhead. In contrast, our proposed 4-bit compression method, with a rank selection of 1/16, only doubles the time, while simultaneously reducing memory usage to approximately 1/7 of the original. This highlights that, compared to CPU-offloading methods, low-bit compression offers a more favorable balance between memory usage and time consumption.

Table 7: The optimizer memory usage (GB) of different methods.

| Model Size | 130M | 350M | 1.1B | 3B | 5B |
|---|---|---|---|---|---|
| fp32-Muon | 0.41 | 0.94 | 3.88 | 11.81 | 19.97 |
| fp32-Muon-CPU | 0 | 0 | 0 | 0 | 0 |
| 4-bit-Muon-GRASP (rank 1/4) | 0.11 | 0.25 | 1.06 | 3.03 | 5.42 |
| 4-bit-Muon-GRASP (rank 1/16) | 0.06 | 0.19 | 0.81 | 2.07 | 3.79 |
| 4-bit-Muon-GRASP (rank 1/64) | 0.05 | 0.16 | 0.74 | 1.92 | 2.86 |

## B.2 DETAILED BREAKDOWN OF OPTIMIZER TIME USAGE

Table 8: Time (ms) of different optimizer logic.

| Model size (rank) | 130M(1/4) | 130M(1/16) | 130M(1/64) | 350M(1/4) | 350M(1/16) | 350M(1/64) | 1.1B(1/4) | 1.1B(1/16) | 1.1B(1/64) |
|---|---|---|---|---|---|---|---|---|---|
| Power iteration | 134 | 46 | 32 | 410 | 129 | 58 | 1010 | 242 | 94 |
| NS iteration | 98 | 98 | 98 | 224 | 224 | 224 | 378 | 378 | 378 |
| param update | 12 | 12 | 12 | 22 | 22 | 22 | 35 | 35 | 35 |

Here, we provide a detailed breakdown of the cost of power iteration versus the rest of the optimizer logic and the results are shown in Tab 8. We find that when the rank is selected as 1/16, the time required for Power Iteration is less than that of the NS iteration. However, when the rank is larger (1/4), the QR decomposition in Power Iteration becomes relatively time-consuming and exceeds the time required for the NS iteration. This pattern holds across different model sizes. In the overall training steps, the forward and backward processes account for the majority of the time, with the increase in optimizer time (0.2-0.6 seconds) considered a minimal overhead.

## B.3 MORE ABLATION RESULTS OF DIFFERENT RANKS.

Table 9: The optimizer memory usage (GB) of different ranks.

| Model Size | 130M | 350M | 1.1B | 3B | 5B |
|---|---|---|---|---|---|
| 4-bit-Muon-GRASP (rank 1/4) | 0.11 | 0.25 | 1.06 | 3.03 | 5.42 |
| 4-bit-Muon-GRASP (rank 1/8) | 0.08 | 0.23 | 0.92 | 2.57 | 4.62 |
| 4-bit-Muon-GRASP (rank 1/16) | 0.06 | 0.19 | 0.81 | 2.07 | 3.79 |
| 4-bit-Muon-GRASP (rank 1/32) | 0.05 | 0.17 | 0.78 | 1.99 | 3.32 |
| 4-bit-Muon-GRASP (rank 1/64) | 0.05 | 0.16 | 0.74 | 1.92 | 2.86 |

Here, we also provide the memory usage and time of different rank chosen, and the results are shown in Tab. 9 and Tab. 10. To highlight the contrast more clearly, here we focus only on the optimizer itself. These results reveal a trade-off: as the rank increases, model performance improves, but this comes at the cost of increased time consumption and memory usage. Consequently, in practical training, we propose selecting the highest possible rank that remains within the acceptable limits of memory and time usage to achieve a good performance. Furthermore, we have included additional visualizations of the singular value distributions in Appendix B.5, which reveal that the distributions exhibit similar patterns across different model sizes. Therefore, the rank selected for smaller models can also be transferred to larger models.

## B.4 DATA FORMATS AND LEARNING RATES

Here, we compare the impact of FP4 and INT4 data formats on quantization. The implementation of INT4 is straightforward: after dividing by the normalization scale, we apply the *round()* function. For FP4 representation, we adopt the E2M1 format as defined in prior studies (Rouhani et al., 2023), which includes the following values:

$$\{-6, -4, -3, -2, -1.5, -1, -0.5, 0, 0.5, 1, 1.5, 2, 3, 4, 6\} \tag{14}$$

Table 10: The optimizer updating time(s) of different ranks.

| Model Size | 130M | 350M | 1.1B | 3B | 5B |
|---|---|---|---|---|---|
| 4-bit-Muon-GRASP (rank 1/4) | 0.55 | 0.91 | 1.82 | 2.85 | 4.21 |
| 4-bit-Muon-GRASP (rank 1/8) | 0.46 | 0.72 | 1.30 | 1.94 | 2.72 |
| 4-bit-Muon-GRASP (rank 1/16) | 0.41 | 0.61 | 0.96 | 1.21 | 1.93 |
| 4-bit-Muon-GRASP (rank 1/32) | 0.40 | 0.56 | 0.83 | 1.04 | 1.60 |
| 4-bit-Muon-GRASP (rank 1/64) | 0.39 | 0.54 | 0.76 | 0.88 | 1.43 |

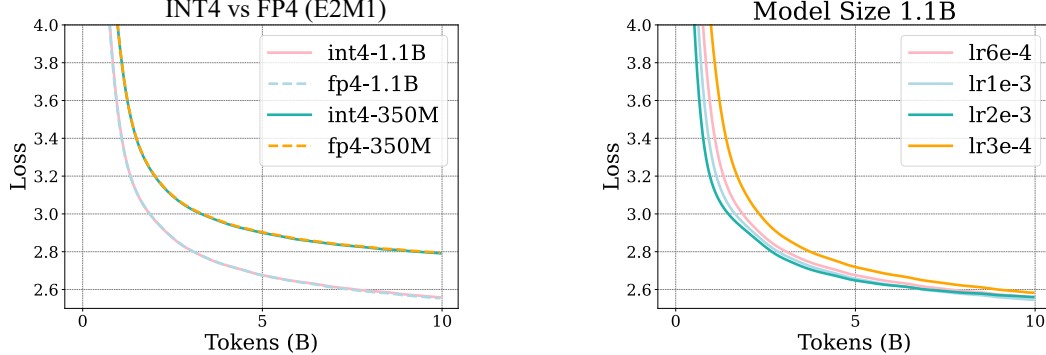

Figure 9: Left: Comparision of INT4 and FP4 data format. Right: Different learning rate.

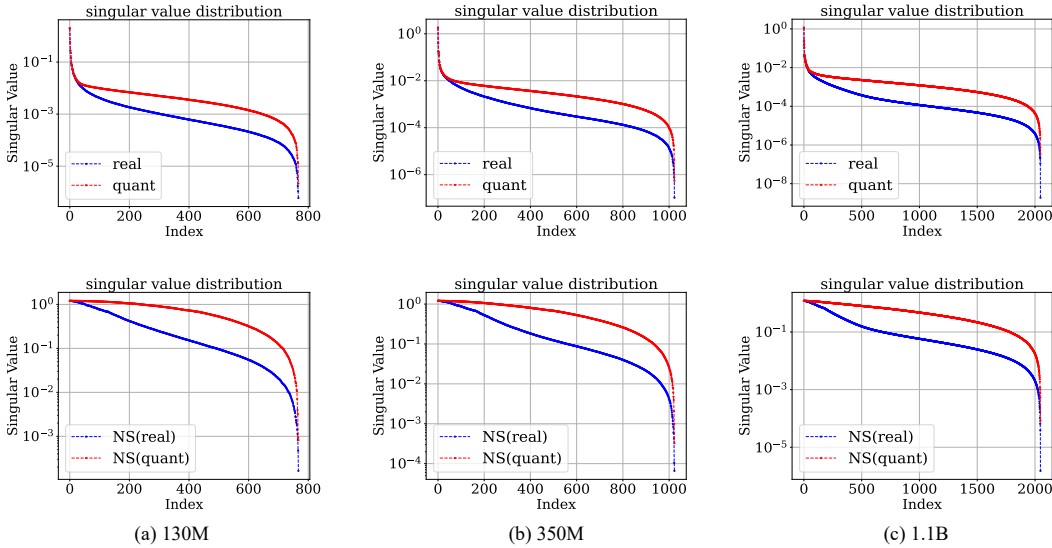

(a) 130M      (b) 350M      (c) 1.1B

Figure 10: Visualization of momentum in *transformer.layers.7.attn.o_proj* in different size of LLaMA model.

Following Wang et al. (2025), we implement a look-up table for FP4 quantization within a Triton kernel. Quantization functions typically involve element-wise operations on large datasets, which can be parallelized to leverage the highly parallel computing power of GPUs. Fig. 9 (left) shows that the training curves for both data formats are nearly identical, but the look-up process in FP4 introduces a slight efficiency overhead.

We also compare the effect of different learning rates on the convergence of training with 4-bit-Muon-GRASP optimizer. We select learning rates from the set {3e-4, 6e-4, 1e-3, 3e-3}. Figure 9 (right) shows that 4-bit-Muon-GRASP converges to similar levels across different learning rates.

### B.5 Singular value distribution of the moment matrices for models of different sizes.

Here we present a single representative visualization illustrating the singular value distribution of the moment matrices across different model scales, see Fig 10.

## C Broader Discussion of low-bit-Muon Beyond LLMs

Beyond LLM pretraining, low-bit optimizers can benefit a wide range of memory-constrained learning settings (Qin et al., 2025; Lu et al., 2024). In particular, Muon has already demonstrated strong empirical performance beyond language modeling, including in computer vision and other large-scale training regimes, highlighting the generality of optimizer-level improvements. In combinatorial optimization, where neural solvers often operate on large graphs and long trajectories (Wang et al., 2023; Guo et al., 2024; Zhang et al., 2024a; Li et al., 2025a; Zheng et al., 2024), reducing optimizer-state precision directly lowers the training memory footprint and enables scaling to larger problem instances. Similar benefits arise in bioinformatics (Wu et al., 2024a; Bian et al., 2026), where high-dimensional biological data and limited hardware budgets make memory efficiency a practical bottleneck.

Similarly, in generative modeling—including diffusion models, autoregressive transformers (Li et al., 2025b;b), and graph-based molecule generators—training (Yang et al., 2024b; Wu et al., 2024b; Yang et al., 2025b) is frequently dominated not only by model parameters but also by optimizer states. Low-bit optimizers can therefore substantially reduce the training memory footprint, facilitating scaling to higher resolutions, longer generation horizons, or larger latent spaces. This is particularly beneficial for scientific generative tasks such as molecular and materials design, where models must process complex graph or 3D structural representations. More broadly, low-bit optimizers provide an orthogonal direction to existing efficiency techniques such as parameter-efficient training and model quantization, targeting the often-overlooked memory cost of optimizer states and enabling scalable learning across a wide range of domains beyond LLMs.

