# OpenReview forum: "Achieving low-bit Muon through subspace preservation and grid quantization"
_ICLR.cc/2026/Conference — ICLR 2026 Poster_

### Official Review · Reviewer_n2Ba · 2025-10-31

**Soundness:** 3
**Presentation:** 3
**Contribution:** 3
**Rating:** 6
**Confidence:** 4

**Summary:**

This paper addresses the challenge of reducing the memory footprint of the Muon optimizer, which is already a memory-efficient alternative to AdamW as it only stores the first moment. The authors investigate applying 4-bit quantization to Muon's optimizer states, a non-trivial task as they demonstrate that Muon's core orthogonalization step (the Newton-Schulz iteration) significantly amplifies quantization errors.

The paper identifies that this error amplification originates primarily from the top singular subspace of the moment matrix. To solve this, the authors propose 4-bit-Muon-GRASP (GRid And Subspace Preserving).

Experiments on LLaMA pre-training (up to 1.1B) and fine-tuning of 7B models show that 4-bit-Muon-GRASP achieves accuracy comparable to its full-precision counterpart, while reducing total training memory consumption by up to 28% compared to fp32-Muon.

**Strengths:**

* The paper identifies the Newton-Schulz (NS) iteration as an amplifier for quantization error, and successfully pinpoints the source of the error amplification to the top singular subspace.
* The proposed 4-bit-Muon-GRASP method is a logical and novel response to the identified problem. It directly treats the "sensitive" top subspace with higher precision (8-bit) while aggressively compressing the "stable" residual (4-bit).
* The introduction of grid quantization to handle outliers in both row and column dimensions is a simple but effective technique that outperforms standard group quantization.
* The paper demonstrates that 4-bit-Muon-GRASP achieves its goal, matching the performance of the full-precision fp32-Muon optimizer on both pre-training (130M, 350M, 1.1B models) and challenging fine-tuning tasks (7B models).

**Weaknesses:**

1.  **Limited Pre-training Scale:** The pre-training experiments are limited to models up to 1.1B parameters. While fine-tuning is performed on 7B models, the most critical need for optimizer memory savings is during large-scale pre-training. The memory gains are also less substantial at smaller scales. Figure 6 compellingly shows projected memory savings for 3B and 5B models, but lacks the corresponding training loss curves to prove that GRASP holds up at that scale. The authors acknowledge this limitation.
2.  **Baselines for 4-bit Muon:** The main low-bit baseline, "4-bit-Muon-base", is a naive implementation. The paper's argument would be stronger if it discussed why other, more sophisticated 4-bit quantization techniques would *also* fail. Is the NS-iteration-based amplification a fundamental blocker for *all* previous quantization methods?
3.  **Computational Overhead of Subspace Preservation:** The method adds a Power Iteration step at every iteration to compute the top singular vectors. While Table 3 shows the total step time overhead is "minimal", a more detailed breakdown of the cost of this step versus the rest of the optimizer logic (including the NS iterations) would be beneficial.
4.  **Clarity of Algorithm 1:** Algorithm 1 is slightly difficult to parse. For example, the `PowerIter` function also uses `Orthogonalize(P)` (line 3) which is implemented via QR decomposition, but the main text refers to this as "column normalization", which is a bit ambiguous.

**Questions:**

1.  The main experiments use a rank of 1/16 for the top subspace. The ablation in Fig. 7a shows that rank matters, and 1/16 is a clear compromise between 1/8 and 1/32. How sensitive is this hyperparameter? Does it need to be re-tuned for different model sizes or architectures, or is 1/16 generally robust?
2.  The ablation in Fig. 7c shows that discarding the residual singular space leads to poor performance. This is a key finding. Does this imply that the primary insight is that Muon (due to its orthogonalization) *cannot* be approximated by a simple low-rank matrix, and *must* retain the full-rank information, even if the residual part is heavily quantized?
3.  Regarding the computational overhead, how does the cost of the single Power Iteration step compare to the cost of the 5-step NS iteration? A more granular profiling would help in understanding the trade-offs.

---

> ### Author Response · Authors · 2025-11-21
> **Rebuttal (1/2)**
>
> Thank you for acknowledging our work and your suggestions and questions are really helpful and valuable. Below is our detailed response.
>
> > **W2**: Baselines for 4-bit Muon: The main low-bit baseline, "4-bit-Muon-base", is a naive implementation. The paper's argument would be stronger if it discussed why other, more sophisticated 4-bit quantization techniques would also fail. Is the NS-iteration-based amplification a fundamental blocker for all previous quantization methods?
>
> Thanks. First, we apologize for any misunderstanding caused by the use of the term "naive." We would like to humbly clarify that the "4-bit-Muon-base" is not as simple as it may appear. It utilizes the quantization technique proposed in [1], which incorporates group quantization and dynamic exponent mapping. To the best of our knowledge, [1] represents the latest 4-bit advancement on AdamW, achieving strong results. Additionally, group quantization, compared to simpler methods like tensor quantization or block quantization, is already a more sophisticated and finer-grained approach.
>
> As this is the first exploration into the quantization and compression of the Muon optimizer, we are sorry that we currently do not have a stronger baseline for comparison.
>
> [1] Li, Bingrui, et al. “Memory efficient optimizers with 4-bit states.”
>
>
> > **W3**: Computational Overhead of Subspace Preservation: The method adds a Power Iteration step at every iteration to compute the top singular vectors. While Table 3 shows the total step time overhead is "minimal", a more detailed breakdown of the cost of this step versus the rest of the optimizer logic (including the NS iterations) would be beneficial.
>
> **Re.Table 1. Time (ms) of different optimizer logic.**
>
> | Model size (rank) | 130M(1/4) | 130M(1/16) | 130M(1/64) | 350M(1/4) | 350M(1/16) | 350M(1/64) | 1.1B(1/4) | 1.1B(1/16) | 1.1B(1/64) |
> | ----------------- | --------- | ---------- | ---------- | --------- | ---------- | ---------- | --------- | ---------- | ---------- |
> | Power iteration   | 134       | 46         | 32         | 410       | 129        | 58         | 1010      | 242        | 94         |
> | NS iteration      | 98        | 98         | 98         | 224       | 224        | 224        | 378       | 378        | 378        |
> | param update      | 12        | 12         | 12         | 22        | 22         | 22         | 35        | 35         | 35         |
>
> Here, we provide a detailed breakdown of the cost of power iteration versus the rest of the optimizer logic in Re.Table1. We find that when the rank is selected as 1/16, the time required for Power Iteration is less than that of the NS iteration. However, when the rank is larger (1/4), the QR decomposition in Power Iteration becomes relatively time-consuming and exceeds the time required for the NS iteration. This pattern holds across different model sizes. In the overall training steps, the forward and backward processes account for the majority of the time, with the increase in optimizer time (0.2-0.6 seconds) considered a minimal overhead.
>
>
> > **W4**: Clarity of Algorithm 1: Algorithm 1 is slightly difficult to parse. For example, the PowerIter function also uses Orthogonalize(P) (line 3), which is implemented via QR decomposition, but the main text refers to this as "column normalization", which is a bit ambiguous.
>
> We apologize for any ambiguity in our wording, the "column normalization" mentioned in the main text refers to Line 8 of the algorithm. We have revised our manuscript to make it clearer and thank you for pointing them out.

---

> ### Author Response · Authors · 2025-11-21
> **Rebuttal (2/2)**
>
> > **Q1**: Rank 1/16 is a clear compromise between 1/8 and 1/32. How sensitive is this hyperparameter? Does it need to be re-tuned for different model sizes or architectures, or is 1/16 generally robust?
>
>
> **Re.Table 2. The optimizer updating time(s) of different methods.**
>
> | Model Size                   | 130M | 350M | 1.1B | 3B   | 5B   |
> | ---------------------------- | ---- | ---- | ---- | ---- | ---- |
> | 4-bit-Muon-GRASP (rank 1/4)  | 0.55 | 0.91 | 1.82 | 2.85 | 4.21 |
> | 4-bit-Muon-GRASP (rank 1/8)  | 0.46 | 0.72 | 1.30 | 1.94 | 2.72 |
> | 4-bit-Muon-GRASP (rank 1/16) | 0.41 | 0.61 | 0.96 | 1.21 | 1.93 |
> | 4-bit-Muon-GRASP (rank 1/32) | 0.40 | 0.56 | 0.83 | 1.04 | 1.60 |
> | 4-bit-Muon-GRASP (rank 1/64) | 0.39 | 0.54 | 0.76 | 0.88 | 1.43 |
>
>
>
> **Re.Table 3. The optimizer memory usage (GB) of different methods.**
>
> | Model Size                   | 130M | 350M | 1.1B | 3B   | 5B   |
> | ---------------------------- | ---- | ---- | ---- | ---- | ---- |
> | 4-bit-Muon-GRASP (rank 1/4)  | 0.11 | 0.25 | 1.06 | 3.03 | 5.42 |
> | 4-bit-Muon-GRASP (rank 1/8)  | 0.08 | 0.23 | 0.92 | 2.57 | 4.62 |
> | 4-bit-Muon-GRASP (rank 1/16) | 0.06 | 0.19 | 0.81 | 2.07 | 3.79 |
> | 4-bit-Muon-GRASP (rank 1/32) | 0.05 | 0.17 | 0.78 | 1.99 | 3.32 |
> | 4-bit-Muon-GRASP (rank 1/64) | 0.05 | 0.16 | 0.74 | 1.92 | 2.86 |
>
> Thanks. The rank behaves like a hyperparameter, and selecting an appropriate rank involves a tradeoff: as the rank increases, model performance improves (see Fig.7(a) in the manuscript), but this comes at the cost of increased time consumption and memory usage (see Re.Table 2&3). From these results, we can observe that the rank selection between 1/8 and 1/64 is not sensitive.
>
> In practical training, we propose selecting the highest possible rank that remains within the acceptable limits of memory and time usage to achieve a good performance.
>
> Furthermore, we have included additional visualizations of the singular value distributions in Appendix B.4, which reveal that the distributions exhibit similar patterns across different model sizes. Therefore, the rank selected for smaller models can also be transferred to larger models, which implies 1/16 is a generally robust choice.
>
>
> > **Q2**: Does this imply that the primary insight is that Muon (due to its orthogonalization) cannot be approximated by a simple low-rank matrix, and must retain the full-rank information, even if the residual part is heavily quantized?
>
> Thank you for the valuable question. Yes, the NS iteration amplifies all singular values, causing even originally small values to be magnified to an extent that cannot be overlooked. This makes it challenging to apply low-rank approximation to Muon. As you mentioned, our ablation study also demonstrates this key finding, indicating that the information in the residual part must also be preserved, which we deal with 4-bit compression.
>
>
> > **Q3**: How does the cost of the single Power Iteration step compare to the cost of the 5-step NS iteration? A more granular profiling would help in understanding the trade-offs.
>
>
> **Re.Table 4. Time(ms) comparison between single Power Iteration and a 5-step NS iteration.**
>
> | Model size (rank) | 130M(1/4) | 130M(1/16) | 130M(1/64) | 350M(1/4) | 350M(1/16) | 350M(1/64) | 1.1B(1/4)    | 1.1B(1/16)   | 1.1B(1/64)   |
> | ----------------- | --------- | ---------- | ---------- | --------- | ---------- | ---------- | --- | --- | --- |
> | Power Iter time   |     134      |      46      |      32      |       410    |     129   |   58   | 1010    |  242   |    94 |
> | NS time           |         98  |      98      |     98      |     224      |      224      |           224 |    378 |   378  |  378   |
>
>
> **Q3 is similar with W3**. Thank you for the valuable suggestions. Here, we provide a comparison between a single Power Iteration step and a 5-step NS iteration in Re.Table 4, across different model sizes and rank selections. We find that when the rank is selected as 1/16, the time required for Power Iteration is less than that of the NS iteration. However, when the rank is larger (1/4), the QR decomposition in Power Iteration becomes relatively time-consuming and exceeds the time required for the NS iteration. This pattern holds across different model sizes.
>
> ---
> We hope our detailed response could answer your questions and address your concerns, looking forward to receiving your feedback soon.

---

> > ### Author Response · Authors · 2025-11-26
> > **Looking forward your feedback**
> >
> > Dear Reviewer n2Ba,
> >
> > We hope that our responses adequately address your concerns. As the deadline of this discussion phase is approaching, we warmly welcome further discussion regarding any additional concerns that you may have, we are looking forward to receiving your feedback soon.
> >
> > Thank you for the time and appreciation that you have dedicated to our work.
> >
> > Best regards,
> >
> > Authors of submission 5013

---

> > > ### Comment · Reviewer_n2Ba · 2025-11-28
> > >
> > > Thanks for the detailed response, my concerns has been addressed, I would maintain my current positive review score.

---

> > > > ### Author Response · Authors · 2025-11-28
> > > >
> > > > Dear Reviewer n2Ba:
> > > >
> > > > Thank you for your response! We’re glad our rebuttal addressed your concerns, and we appreciate your decision to maintain a positive evaluation.
> > > >
> > > > Thank you for the time and appreciation that you have dedicated to our work.
> > > >
> > > > Best regards,
> > > >
> > > > Authors of submission 5013

---

### Official Review · Reviewer_r7tv · 2025-10-31

**Soundness:** 3
**Presentation:** 3
**Contribution:** 3
**Rating:** 8
**Confidence:** 3

**Summary:**

The paper proposes an approach to quantizer momentum state in Muon optimizer to 4-bit. The approach involves preserving the singular values of momentum in 8-bit while the residual components are quantized to 4-bit. The idea of subspace preserving mixed precision quantization is intuitive and the grid based quantization scheme seem to work better than group based quantization.
To my knowledge, this is the first work to perform 4-bit quantization of Muon optimizer so there aren't any related baselines to compare with. The evaluations in this work are thorough and the ablations are useful to motivate the approach.

**Strengths:**

1. The approach seems to work compared to baseline approach of 4-bit Muon quantization without subspace preservation.

**Weaknesses:**

1. Need to show results on multiple seeds.
2. Code is not provided.

**Questions:**

1. Do the authors have any insight on the performance of quantized 4-bit Muon v/s 4-bit AdamW optimizer?
2. Is muon optimizer more difficult to quantize than adamw optimizer?

---

> ### Author Response · Authors · 2025-11-21
> **Rebuttal**
>
> Thank you for acknowledging our work and your rating has provided us with great encouragement!!  Below is our detailed response.
>
> > **W1**: Need to show results on multiple seeds.
>
> Thank you for the suggestion. In the revised version, we report the results evaluated using multiple random seeds (1234, 12345, 1337). These random seeds are based on the ones used in https://github.com/karpathy/nanoGPT/blob/master/train.py and https://github.com/EleutherAI/lm-evaluation-harness/blob/main/lm_eval/evaluator.py.
>
> > **W2**: Code is not provided.
>
> Thank you. We provide the core code for the 4-bit-Muon-GRASP optimizer in an Anonymous Repo, please see link below. The whole training script will be made public upon acceptance. Thanks for your understanding.
>
>
> > **Q1**: Do the authors have any insight on the performance of quantized 4-bit Muon v/s 4-bit AdamW optimizer?
>
> Thanks. The 4-bit AdamW and 4-bit Muon optimizers are both designed to approximate their full-precision counterparts. Therefore, the comparison between the quantized 4-bit Muon and 4-bit AdamW essentially boils down to a comparison between Muon and AdamW themselves. Several recent studies[1][2] have demonstrated that the Muon optimizer outperforms AdamW to some extent. As a result, it follows that a lossless 4-bit Muon will generally yield superior performance compared to a lossless 4-bit AdamW.
>
> [1] Liu, Jingyuan, et al. "Muon is scalable for LLM training."
>
> [2] Wen, Kaiyue, et al. "Fantastic pretraining optimizers and where to find them.
>
> > **Q2**: Is muon optimizer more difficult to quantize than adamw optimizer?
>
> Thank you. The Muon optimizer only contains the first moment, while the AdamW optimizer includes both the first and second moments in its optimizer states.
>
> For the first moment, AdamW directly uses it for updates in an element-wise fashion, whereas in Muon, it undergoes an orthogonalization step, which is achieved through Newton-Schulz (NS) iterations. We found that NS iterations amplify quantization error, as discussed in Section 3.1. Regarding the second moment in AdamW, several works have already focused on compressing it [3][4].
>
> Therefore, the Muon optimizer is more difficult to quantize than the AdamW optimizer. To the best of our knowledge, the compression of Muon has not been attempted before, and we are the first to explore this.
>
>
> [3] Li, Bingrui, et al. "Memory efficient optimizers with 4-bit states."
>
> [4] Zhang, Yushun, et al. "Adam-mini: Use fewer learning rates to gain more."
>
> ---
> We hope our response could answer your questions and address your concerns, looking forward to receiving your feedback soon.

---

> > ### Author Response · Authors · 2025-11-26
> > **Looking forward your feedback**
> >
> > Dear Reviewer r7tv,
> >
> > We hope that our responses adequately address your concerns. As the deadline of this discussion phase is approaching, we are looking forward to receiving your feedback soon.
> >
> > Thank you for the time and appreciation that you have dedicated to our work.
> >
> > Best regards,
> >
> > Authors of submission 5013

---

> ### Author Response · Authors · 2025-11-21
> **Anonymous Link**
>
> https://anonymous.4open.science/r/ICLR-5013-E07E

---

### Official Review · Reviewer_TV6R · 2025-11-01

**Soundness:** 2
**Presentation:** 3
**Contribution:** 2
**Rating:** 4
**Confidence:** 3

**Summary:**

The authors propose a variant of Muon, where the optimization state M is quantized into a low-bit version. By observing that the singular vectors with large singular values will have large quantization errors after NS, the authors propose to use 8-bit quantization for the low rank matrix associated with the singular vectors with large singular values and to use 4-bit quantization for the residual part. To further decrease quantization error, the authors propose to use grid quantization. The experimental results show that the proposed low-bit Muon achieves similar performance to the 32-bit version, while costing less memory.

**Strengths:**

1. The authors identify that after NS iteration, the relative error of the matrix associated with the singular vectors with large singular values will be amplified a lot.

2. Based on the observation, the authors propose an algorithm that uses different precision for different classes of singular vectors.

3. To further decrease the quantization error, the authors propose grid quantization method that scales the matrix according to the maximum absolute value in each row and each column.

4. The experimental results show a similar performance between the proposed algorithm and the 32-bit Muon.

**Weaknesses:**

1. Why do the authors use a single iteration of the power method? Is it enough for the algorithm to identify the top singular vector? Will it affect the performance?

2. The overall time seems to be more than the 32-bit version. How does the time compare to the CPU-offload muon, which costs 0 memory on GPU but costs more time than the 32-bit version, especially for large models and large k (the rank of Q).

3. The selection of rank will affect the performance even for 350M model. How to choose the rank for even a large model?

**Questions:**

See weaknesses.

---

> ### Author Response · Authors · 2025-11-21
> **Rebuttal (1/2)**
>
> Thanks for your review and all valuable questions, below is our detailed response.
>
>
> > **W1**: Why do the authors use a single iteration of the power method? Is it enough for the algorithm to identify the top singular vector? Will it affect the performance?
>
>
> Thanks. In this work, we use single-step power iteration to avoid the need of Singular Value Decomposition (SVD) for the top singular space $M_{top}$. This single iteration has also been employed in other works for approximation, such as PowerSGD[1] and Dion[2]. The number of iterations here involves a tradeoff, as more iterations require longer computation time but can improve approximation accuracy.
>
> Meanwhile, it should be noted that we reuse previously computed matrix approximations to initialize the power iteration algorithm. At time $t$, we perform column normalization on $R_{t-1}$ to obtain $Q_t$, and use $Q_t$ to perform a single step of power iteration. Since $M_t$ and $M_{t-1}$ exhibit a certain degree of similarity, we can benefit from reusing $R_{t-1}$ as the starting point. In this way, we can minimize the time overhead introduced by power iteration (only single step), and we believe it is sufficient to approximate the top singular vector.
>
>
> Moreover, we conduct an experiment to evaluate the approximation accuracy of Power Iteration with different steps, the corresponding results are shown in Figure 8 in the revised manuscript. The results show that the error gap between the 1-step iteration and the multi-step iterations is minimal, and the approximation error of the 1-step power method already reaches as low as 0.01, thus single step iteration is enough for the algorithm to identify the top singular vector.
>
> [1] Vogels, Thijs, et, al. "PowerSGD: Practical low-rank gradient compression for distributed optimization."
>
> [2] Ahn, Kwangjun, et al. "Dion: Distributed orthonormalized updates."
>
>
> > **W2**: The overall time seems to be more than the 32-bit version. How does the time compare to the CPU-offload muon, which costs 0 memory on GPU but costs more time than the 32-bit version, especially for large models and large k (the rank of Q).
>
> Thanks. To highlight the contrast more clearly, here we focus only on the optimizer itself. Below is the comparison of **optimizer updating time** and **optimizer memory usage** of different methods (32-bit-Muon, 32-bit-Muon-CPU, 4-bit-Muon-GRASP).
>
>
> **Re.Table 1. The optimizer updating time(s) of different methods.**
>
> | Model Size                   | 130M | 350M | 1.1B | 3B    | 5B    |
> | ---------------------------- | ---- | ---- | ---- | ----- | ----- |
> | fp32-Muon                    | 0.18 | 0.25 | 0.39 | 0.55  | 0.75  |
> | fp32-Muon-CPU                | 0.73 | 0.99 | 3.89 | 11.32 | 17.93 |
> | 4-bit-Muon-GRASP (rank 1/4)  | 0.55 | 0.91 | 1.82 | 2.85  | 4.21  |
> | 4-bit-Muon-GRASP (rank 1/16) | 0.41 | 0.61 | 0.96 | 1.21  | 1.93  |
> | 4-bit-Muon-GRASP (rank 1/64) | 0.39 | 0.54 | 0.76 | 0.88  | 1.43  |
>
> **Re.Table 2. The optimizer memory usage (GB) of different methods.**
>
> | Model Size                   | 130M | 350M | 1.1B | 3B    | 5B    |
> | ---------------------------- | ---- | ---- | ---- | ----- | ----- |
> | fp32-Muon                    | 0.41 | 0.94 | 3.88 | 11.81 | 19.97 |
> | fp32-Muon-CPU                | 0    | 0    | 0    | 0     | 0     |
> | 4-bit-Muon-GRASP (rank 1/4)  | 0.11 | 0.25 | 1.06 | 3.03  | 5.42  |
> | 4-bit-Muon-GRASP (rank 1/16) | 0.06 | 0.19 | 0.81 | 2.07  | 3.79  |
> | 4-bit-Muon-GRASP (rank 1/64) | 0.05 | 0.16 | 0.74 | 1.92  | 2.86  |
>
> The above results demonstrate that the optimizer step time for CPU-offloaded Muon increases significantly, especially as the model size grows. For a 5B model, the optimizer time becomes 20 times longer, introducing a substantial overhead. In contrast, our proposed 4-bit compression method, with a rank selection of 1/16, only doubles the time, while simultaneously reducing memory usage to approximately 1/7 of the original. This highlights that, compared to CPU-offloading methods, low-bit compression offers a more favorable balance between memory usage and time consumption.

---

> ### Author Response · Authors · 2025-11-21
> **Rebuttal (2/2)**
>
> > **W3**: The selection of rank will affect the performance even for 350M model. How to choose the rank for even a large model?
>
> Thanks. The rank behaves like a hyperparameter, and selecting an appropriate rank involves a tradeoff: as the rank increases, model performance improves, but this comes at the cost of increased time consumption and memory usage (see Re.Table 1&2 above). Consequently, in practical training, we propose selecting the highest possible rank that remains within the acceptable limits of memory and time usage to achieve a good performance.
>
> Furthermore, we have included additional visualizations of the singular value distributions in Appendix B.4, which reveal that the distributions exhibit similar patterns across different model sizes. Therefore, the rank selected for smaller models can also be transferred to larger models.
>
>
> ---
>
> We have added all the above descriptions and experiments into the revised manuscript and marked as blue.
>
> We hope our responses and modifications could ease your concerns. If you have any other questions, we are glad to provide further responses. We would sincerely appreciate it if you could reconsider your rating and wish to receive your feedback soon.

---

> > ### Author Response · Authors · 2025-11-26
> > **Looking forward your feedback**
> >
> > Dear Reviewer TV6R,
> >
> > We hope that our responses adequately address your concerns. As the deadline of this discussion phase is approaching, we warmly welcome further discussion regarding any additional concerns that you may have, and we sincerely hope you can reconsider the rating accordingly.
> >
> > Thank you for the time and appreciation that you have dedicated to our work.
> >
> > Best regards,
> >
> > Authors of submission 5013

---

> > > ### Comment · Reviewer_TV6R · 2025-11-28
> > >
> > > Thanks for the authors' response. From the additional experiments, the quantization of Muon can actually save time and memory compared to the CPU-offload version and the FP32 version, respectively. Thus, I will increase the score to 6.

---

> > > > ### Author Response · Authors · 2025-11-28
> > > >
> > > > Dear Reviewer TV6R:
> > > >
> > > > Thank you for carefully considering our additional experiments and for your constructive feedback. We’re glad the results have addressed your concerns, and we appreciate your decision to raise the score to 6. Thanks again!
> > > >
> > > > Best regards,
> > > >
> > > > Authors of submission 5013

---

### Official Review · Reviewer_H32P · 2025-11-02

**Soundness:** 3
**Presentation:** 3
**Contribution:** 3
**Rating:** 6
**Confidence:** 4

**Summary:**

The authors propose 4-bit-Muon-GRASP, a novel approach for reducing storage requirements for Muon’s momentum state. This is important for practitioners training in memory-constrained scenarios. The authors find that directly quantizing Muon’s momentum to 4 bits leads to large errors relative to an FP32 baseline after Newton Shulz (NS) orthogonalization is applied. The authors show that these errors are mainly caused by quantizing the top singular space. Therefore, they break the quantization problem into 2 parts to reduce errors caused by NS: (1) they quantize the top singular subspace to 8 bits, and (2) they quantize the remainder to 2 bits. In their experiments spanning language model pre-training and finetuning, the authors achieve this naturally through power iteration. In their experiments, the authors show that 4-bit-Muon-GRASP nearly matched the convergence of the FP32 baseline.

**Strengths:**

- The paper is easy to understand and follows a coherent story.
- The experiments provide a good evaluation of the proposed method and are well-executed.
- The authors provide the timing and memory complexity of their method.
- The approach is intuitive.

**Weaknesses:**

- Due to grid quantization’s reliance on both the column and row scales, the dequantization step is not compatible with optimizer state sharding(zero-1/fsdp) out of the box, despite the combination of these methods being relevant for low-memory training.
- While 8bit-Muon is included in Figure 6 (memory usage), I miss a comparison to it in most other performance-related experiments. It seems like 8-bit Muon is a much simpler approach than 4-bit-Muon-GRASP, so it would be beneficial for practitioners to understand how it performs.
- I miss a description of how hyperparameters were chosen? Were the optimal values an interior point of all the values searched? This is crucial to have sound results.

**Questions:**

- Why are the singular values all below 1 for the NS(real) in Figure 1 (d)? Have you swapped the labels by accident? If not, the plot reads like the quantized NS is closer to UV^T (singular values 1) than real NS.
- I’m not sure exactly what rank refers to in Table 1.  I assume you are referring to the number of dimensions in the top singular subspace, but the writing does not make this clear.
- In a distributed optimization setting, [1] finds that Muon trajectories can be quantized to 2 bits. It could be interesting to discuss how this relates to your work.
- What does official choice mean in Figure 2?


**Nit-picks/suggestions:**
- Figure 2:  The y-axis could show “Quantization Error RE(A,B)” without needing a title
- line 244 missing citation
- equation 8, tensors should be in mxk and nxk

[1][MuLoCo: Muon is a practical inner optimizer for DiLoCo]

---

> ### Author Response · Authors · 2025-11-21
> **Rebuttal (1/2)**
>
> Thank you for acknowledging our work. Your rating has provided us with great encouragement, and your detailed suggestions are really helpful. Below is our response.
>
>
> > **W1**. Due to grid quantization’s reliance on both the column and row scales, the dequantization step is not compatible with optimizer state sharding(zero-1/fsdp) out of the box, despite the combination of these methods being relevant for low-memory training.
>
> Thanks. We would like to humbly point out that the grid quantization we propose in this paper is based on group quantization. This means that we first divide the moment matrix along the columns and rows with group size $s$, resulting in several blocks in shape **s*s** (see Fig.3 right for reference). In our setup, the group size is 128, which will be commonly divisible by both the sharing row and sharing column. Therefore, our grid scales (shape **s*s**) are also contained within the local sharded box. As a result, the quantization and dequantization process in our method is actually compatible with optimizer state sharding techniques, such as Zero-1 and FSDP.
>
>
> > **W2**: While 8bit-Muon is included in Figure 6 (memory usage), I miss a comparison to it in most other performance-related experiments. It seems like 8-bit Muon is a much simpler approach than 4-bit-Muon-GRASP, so it would be beneficial for practitioners to understand how it performs.
>
>
> Thanks for the valuable suggestion. Here, we have added the training curve of 8-bit Muon and the corresponding performance on downstream tasks, please see the revised manuscipt for details.
>
> At the outset of this work, we had already implemented both 8-bit and 4-bit Muon with group quantization and evaluated their performance on smaller models. We found that the direct 8-bit Muon achieved lossless performance, while the direct 4-bit Muon showed performance degradation. This observation led us to take on the more challenging task of pushing the compression to 4-bit.
>
>
> > **W3**: I miss a description of how hyperparameters were chosen? Were the optimal values an interior point of all the values searched? This is crucial to have sound results.
>
> Thanks. We have described the chosen hyperparameters in Appendix A. For each model size, we first tune the learning rate for the fp32 baseline from the set {2e-3, 1e-3, 6e-4, 3e-4}, selecting the best learning rate based on the validation perplexity. We then apply the same learning rate to the 4bit-Muon-base and 4bit-Muon-GRASP models to ensure a fair comparison. The final chosen learning rates are shown in Appendix.A. Thank you for your suggestion, and we have revised the manuscript to provide a clearer explanation.
>
>
> > **Q1**: Why are the singular values all below 1 for the NS(real) in Figure 1 (d)? Have you swapped the labels by accident? If not, the plot reads like the quantized NS is closer to UV^T (singular values 1) than real NS.
>
> Thanks for the question. The figures shown in the manuscript are correct.
> Since the NS iteration performs as $f(x) = ax+bx^3+cx^5$, and we follow the official design, using a = 3.4445, b = −4.7750, and c = 2.0315, this polynomial leads all the singular values below 1.
>
> We know that NS iteration only changes the singular values, while the singular vectors of the matrix remain unchanged. However, quantization can alter both the singular vectors and the singular values. Therefore, although quantized NS brings the singular values closer to 1, this does not necessarily indicate better results.
>
> > **Q2**: I’m not sure exactly what rank refers to in Table 1. I assume you are referring to the number of dimensions in the top singular subspace, but the writing does not make this clear.
>
> Here the rank equals $k$ in this subsection, which is the rank of the top singular space. Thanks for the question, and we have revised the manuscript to make this clearer.

---

> ### Author Response · Authors · 2025-11-21
> **Rebuttal (2/2)**
>
> > **Q3**: In a distributed optimization setting, [1] finds that Muon trajectories can be quantized to 2 bits. It could be interesting to discuss how this relates to your work.
>
> Thanks for the valuable question.
>
> [1] studies the distributed training algorithm DiLoCo. Building on DiLoCo, [1] uses Muon as the inner optimizer, compresses the states to 2 bits to reduce communication, and adds error feedback (EF) for compensation.
> In their work, the performance gap between 2-bit with and without EF is large (Figure 2b therein), suggesting that EF is the key factor in maintaining the performance of 2-bit compression.
>
> In fact, considering the EF terms, the effective number of bits per element is **quantized state (2bit) + EF (16 or 32 bits). By comparison, in our method, we use 5 bits (1 bit for the top singular subspace on average and 4 bits for the residual subspaces), so we effectively achieve lower bit usage.
>
> We also want to highlight that the two methods may not be directly comparable due to the different setups. In DiLoCo, the goal is to reduce communication by compressing the communicated states to ultra-low precision, while memory usage is less of a concern, and high-precision EF terms still need large memory. In contrast, our primary goal is to reduce the GPU memory footprint of the optimizer throughout training. This leads to different challenges and techniques, even though both approaches use quantization. Combining these lines of work, for example, quantizing error feedback terms, may be an interesting direction for future research.
>
> > **Q4**: What does official choice mean in Figure 2?
>
> Thank you. Figure 2 illustrates the quantization error under different degrees of the polynomial and varying numbers of steps in the NS iteration. The "official choice" refers to a polynomial degree of 5 and 5 steps in the NS iteration, which is the setting recommended by the Muon developers [2] and the Moonshot report [3]. Thus, we have referred to it as the official choice. We have revised the manuscript to clarify this further.
>
> [2] Jordan, Keller, et al. "Muon: An optimizer for hidden layers in neural networks" https://kellerjordan.github.io/posts/muon/
>
> [3] Liu, Jingyuan, et al. "Muon is scalable for LLM training."
>
> > **Nit-picks/suggestions**
>
> Thanks for pointing them out. We have modified all of them.
>
> ---
>
> We hope this response could answer your questions and address your concerns, looking forward to receiving your further feedback soon.

---

> > ### Author Response · Authors · 2025-11-26
> > **Looking forward your feedback**
> >
> > Dear Reviewer H32P,
> >
> > We hope that our responses adequately address your concerns. As the deadline of this discussion phase is approaching, we warmly welcome further discussion regarding any additional concerns that you may have.
> >
> > Thank you for the time and appreciation that you have dedicated to our work.
> >
> > Best regards,
> >
> > Authors of submission 5013

---

> > > ### Comment · Reviewer_H32P · 2025-11-27
> > >
> > > Thank you for your detailed response. I will maintain my score and still believe the paper should be accepted.

---

> > > > ### Author Response · Authors · 2025-11-27
> > > >
> > > > Dear reviewer H32P,
> > > >
> > > > Thank you very much for your support and your follow-up comments. We hope that our rebuttal and the revised manuscript have adequately addressed your concerns. If there are any remaining issues, please feel free to let us know. If you feel that your concerns have been resolved, we would be very grateful if you could consider raising your score to reflect a clear inclination toward acceptance.
> > > >
> > > > Best regards,
> > > >
> > > > The authors

---

### Author Response · Authors · 2025-11-21
**General Response by Authors**

We would like to express our gratitude to all the reviewers for dedicating their time and providing valuable comments. They acknowledged that our work is intuitive (H32P, r7tv), well-executed (H32P, n2Ba, TV6R), contributive (H32P, r7tv, n2Ba), effective (H32P, TV6R, n2Ba), and presents a novel approach (H32P, n2Ba).

While the overall feedback from the reviewers is positive, Reviewer ZRXS raised concerns regarding the effectiveness of single-step power iteration, the efficiency comparison with the CPU-offload method, and the selection of the rank. In the following response, we provide detailed answers to each question and comment, addressing them point-by-point. Additionally, we have revised the manuscript based on the reviewers' suggestions, with all revisions and additions clearly highlighted in blue. Before delving into the detailed responses to each comment, we briefly summarize the major changes in the updated version:

+ Further discussion on the design of single-step power iteration and rank selection.

+ Training curves and downstream task results for int8-Muon.

+ Efficiency comparison with the CPU-offload method.

+ A detailed breakdown of optimizer time costs.

+ Rephrasing of other clarifications requested by reviewers.

We deeply appreciate the suggestions to improve this paper. If you have any further questions, please let us know so that we can provide a timely follow-up response.

---

### Author Response · Authors · 2025-12-02
**A Short Summary of Rebuttal**

Dear AC,

We are deeply appreciative of the time and effort you have devoted to reviewing our paper. To assist you in forming recommendation, we provide a concise summary of the author–reviewer discussion, organized by specific reviewer requests.:

Overall, we have received responses from three reviewers (#H32P,#TV6R,#n2Ba), and **all reviewers are now positive about our work.**


1. **Reviewer #H32P (score: 6; comment: maintain positive, believes the paper should be accepted)**

      + **Compatibility with ZeRO-1/FSDP**: We clarified that our method is compatible with optimizer state sharding (ZeRO-1/FSDP), since our grid quantization is applied at the group level.


      + **8bit-Muon and hyperparameters**: We have added results for 8bit-Muon (Fig.5, Tab.2, Tab.3 in manuscript) and provided a more detailed description of hyperparameter settings.



2. **Reviewer #TV6R (score: 4; comment: will raise the score to 6**)

      + **Effectiveness of single-step Power Iteration**: We clarified the rationale for using a single-step Power Iteration and conducted additional experiments (Fig.8 in the manuscript) showing that single-step Power Iteration provides an accurate estimate of the top singular vectors.

      + **Comparison with CPU-offload method**: We added a comparison between our approach and a 32-bit CPU-offload-Muon (Tab.8 and Tab.9 in the manuscript). The results show that low-bit compression achieves a more favorable trade-off between memory usage and time consumption.


      + **Effect of rank selection**: We further reported results for different ranks, including their impact on loss curves, memory usage, and time consumption (Sec. 4.3 in the manuscript).


3. **Reviewer #r7tv (score: 8)**

      + **Results with multiple random seeds**: We conducted experiments with multiple random seeds and updated the corresponding results.

      + **Code availability**: We released the implementation of our 4bit-Muon-GRASP optimizer via an anonymized link.


4. **Reviewer #n2Ba (score: 6; comment: my concerns have been addressed, maintain positive)**

      + **Baselines chosen**: We clarified that the “4-bit-Muon-base” baseline uses the current state-of-the-art quantization algorithm for AdamW. And our work is the first exploration of quantization for the Muon optimizer to the best of our knowledge.

      + **Detailed breakdown of time cost**: We added a detailed breakdown of the cost of Power Iteration versus the rest of the optimizer logic (Tab.10 in the manuscript).

---

We hope this summary facilitates your review of our revised manuscript and rebuttal contents.

---

### Meta-Review · Area_Chair_hfHK · 2026-01-08

**Summary:**

Across the reviews, there is broad agreement that the paper is clearly written and that the main technical insight is sound: Newton–Schulz orthogonalization in Muon can strongly amplify quantization error, and this amplification is concentrated in the top singular subspace. The proposed GRASP approach (preserving the sensitive top subspace at higher precision while quantizing the residual aggressively, combined with grid quantization) is considered intuitive and is supported by thorough experiments and ablations on the reported scales. The rebuttal meaningfully improved the paper by adding missing comparisons and by clarifying the main algorithmic and systems trade-offs.
However, the decision hinges on a small number of higher-impact uncertainties raised by the more detailed reviews: the lack of demonstrated large-scale pretraining convergence (e.g., 3B/5B) despite projected memory savings, limited breadth of strong low-bit baselines for Muon (and therefore limited support for more general claims about what “must” fail), and only partial validation of distributed sharding practicality (argued to be compatible but not empirically shown). One supportive review recommends acceptance but focuses on more general requests (multi-seed, code release, AdamW comparisons) and does not substantially engage with these core uncertainties, so it carries less weight in the decision.

**Reviewer Concerns:**

Addressed by the rebuttal:
- H32P’s request for 8-bit Muon performance comparisons appears to be addressed via added curves/results; the hyperparameter selection procedure is clarified (baseline LR tuning and consistent application for fair comparison); terminology and figure annotations (e.g., rank definition, “official choice”) are clarified and minor issues are corrected.
- TV6R’s primary concerns are addressed with added justification and evidence for one-step power iteration (including an accuracy study), detailed time and memory tables comparing fp32 Muon, CPU-offloaded Muon, and GRASP under different rank settings, and practical guidance on rank selection.
- n2Ba’s request for granular profiling is addressed with a breakdown of power iteration vs NS iteration vs parameter update costs; algorithm clarity issues are acknowledged and revised.

Still outstanding or only partially addressed:
- The most important remaining gap is n2Ba’s concern about large-scale pretraining: the rebuttal does not clearly provide actual 3B/5B pretraining convergence evidence (loss curves/metrics), even though the motivation for optimizer memory reduction is strongest at that scale.
- Baseline breadth remains limited (n2Ba): the authors clarify their 4-bit baseline is not purely naive, but stronger or more diverse low-bit Muon baselines are still missing, so claims about general failure modes beyond the tested baseline remain under-supported.
- H32P’s distributed sharding compatibility concern is plausibly argued via block/group structure, but it remains only partially validated without explicit implementation detail or empirical demonstration under common ZeRO/FSDP settings.
- r7tv’s requests (multi-seed results, code availability, explicit comparison to 4-bit AdamW) would improve robustness and positioning but are secondary relative to the large-scale pretraining and baseline-coverage concerns.

**Reviewer Scores:**

H32P: likely remains 6 (possibly with higher confidence) given the added 8-bit comparisons and clearer hyperparameter reporting, with sharding compatibility still only partially validated.

TV6R: likely increases from 4 to 5 due to the added evidence on one-step power iteration and the clearer time–memory trade-off versus CPU offload, plus rank guidance.

n2Ba: likely remains 6 (or shifts slightly toward 5–6) because profiling and clarity concerns were addressed, but large-scale pretraining convergence evidence and baseline breadth are still not fully resolved.

r7tv: likely remains 8; their concerns are general and do not appear central to their high recommendation.

---

### Decision · Program_Chairs · 2026-01-26

Accept (Poster)